# Single-Port Extraperitoneal vs. Multiport Transperitoneal Robot-Assisted Radical Prostatectomy: A Propensity Score-Matched Analysis

**DOI:** 10.3390/cancers16172994

**Published:** 2024-08-28

**Authors:** Jaya S. Chavali, Adriana M. Pedraza, Nicolas A. Soputro, Roxana Ramos-Carpinteyro, Carter D. Mikesell, Jihad Kaouk

**Affiliations:** Glickman Urological & Kidney Institute, Cleveland Clinic, 9500 Euclid Avenue, Q10, Cleveland, OH 44195, USA

**Keywords:** radical prostatectomy, prostate cancer, robotic surgery, single-port, multi-port

## Abstract

**Simple Summary:**

Following its introduction, the innovative purpose-built Single Port (SP) robotic platform has been utilized for various urological surgical procedures, including for the management of clinically significant prostate cancer. This study sought to compare the different cancer-related and postoperative functional outcomes between the novel SP Extraperitoneal robotic radical prostatectomy (RARP) and the gold-standard Transperitoneal multi-port RARP approaches. The findings from this study can be beneficial to assist clinicians to determine best surgical approach based on the individual patient’s characteristics to ensure the favorable outcomes following RARP.

**Abstract:**

**(1) Background:** Since the introduction of the purpose-built Single Port (SP) robotic platform, there has been an ongoing debate regarding its advantages compared to the established multi-port (MP) system. The goal of this present study is to compare the perioperative, oncological, and functional outcomes of SP Extraperitoneal robotic radical prostatectomy (RARP) versus that of MP Transperitoneal RARP approach at a high-volume tertiary center. **(2) Methods:** Based on a retrospective review of the prospectively maintained IRB-approved database, 925 patients successfully underwent RARP by a single experienced robotic surgeon. A 4:1 propensity-matched analysis based on the baseline prostate cancer International Society of Urological Pathology (ISUP) Grade Group, clinical stage, and preoperative Prostate Specific Antigen (PSA) was performed, which yielded a cohort of 606 patients—485 in the SP EP and 121 in the MP TP approaches. Of note, the SP EP approach also included the traditional Extraperitoneal (n = 259, 53.4%) and the novel Transvesical (TV) approaches (n = 226, 46.6%). **(3) Results:** The overall operative time was slightly longer in the SP cohort, with a mean of 198.9 min compared to 181.5 min for the MP group (*p* < 0.001). There were no intraoperative complications with the MP approach and only one during the SP approach. The SP EP technique demonstrated significant benefits, encompassing reduced intraoperative blood loss (SP 125.1 vs. MP 215.9 mL), shorter length of hospital stay (SP 12.6 vs. MP 31.9 h), reduced opioid use at the time of discharge (SP 14.4% vs. MP 85.1%), and an earlier Foley catheter removal (SP 6 vs. MP 8 days). From an oncological perspective, the rate of positive surgical margins remained comparable across both groups (*p* = 0.84). Regarding functional outcomes, the mean continence rates and Sexual Health Inventory for Men (SHIM) scores were identical between the two groups at 6 weeks, 3 months, and 6 months respectively. **(4) Conclusion:** SP EP RARP demonstrates similar performance to MP TP RARP in terms of oncologic and functional outcomes. However, SP EP RARP offers several advantages in reducing the overall hospital stay, decreasing postoperative pain and hence the overall opioid use, as well as shortening the time to catheter removal, all of which translates to reduced morbidity and facilitates the transition to outpatient surgery.

## 1. Introduction

The traditional multiport (MP) robot-assisted radical prostatectomy (RARP) has been the gold standard approach for almost two decades for clinically localized prostate cancer. Even with multiple publications of an extraperitoneal (EP) approach for MP-RARP, the general technique did not have wider acceptance among surgeons given the difficult ergonomics of operating in a narrow surgical field with multiple robotic arms. Following its introduction in 2018, the purpose-built Single Port (SP) robotic platform (Intuitive Surgical, Sunnyvale, CA, USA) has been increasingly utilized for various approaches to RARP [1]. Compared to the traditional MP platform, the novel SP system remains unique given the narrow profile of the single robotic arm with four instrument drives that can simultaneously accommodate one flexible, three-dimensional high-definition endoscopic camera along with three double-jointed endoscopic instruments. Along with the capacity for floating-docking technique, as facilitated by the purpose-built SP Access Kit (Intuitive Surgical, Sunnyvale, CA, USA), the unique features of the SP allowed for improved maneuverability and ergonomics in smaller and shallow surgical working space, which in turn provided the opportunity for surgeons to regionalize surgical approaches to the relevant target anatomy [2,3,4].

Within the domain of RARP, the utility of the SP robotic platform has been demonstrated for various SP-RARP approaches, including Transperitoneal (TP), Transperineal, and the more regionalized Extraperitoneal (EP) and Transvesical (TV) techniques [1,5,6,7,8]. Of note, the clinical benefits of the regionalized EP approach have also been previously shown in the literature, including enhanced postoperative recovery with shorter length of inpatient stays favoring same-day discharges, reduced postoperative pain and opioid prescription, as well as the reduced risk of major postoperative complication [9,10,11]. Furthermore, the TV approach has also allowed for an expanded surgical indication for RARP, providing the opportunity to pursue the procedure in patients with a hostile abdomen or frozen pelvis, as well as with the patient awake under epidural anesthesia. These can be facilitated given the routine use of supine patient positioning as well as the ability to complete the entire steps of RARP from within the confines of the bladder without any disruption on the surrounding anatomy and without the need for bowel manipulation [12,13].

Acknowledging the expanding repertoire of minimally invasive surgical approaches for radical prostatectomy (RP), this study sought to provide additional insight on the potential differences surrounding the perioperative, oncological, and functional outcomes between the SP EP RARP and the MP TP RARP at a high-volume tertiary referral center.

## 2. Methods

### 2.1. Patient Population

We performed a single-center retrospective review of the Institutional Review Board (IRB) approved RP database. We studied the prospectively collected medical records of patients who underwent RARP between January 2015 and October 2023 using the SP or MP surgical platform. The RARP procedures were performed by a single experienced robotic surgeon, using either the da Vinci Xi or SP robotic surgical systems (Intuitive Surgical, Sunnyvale, CA, USA) using techniques that have been described previously [7,8,14]. We routinely performed standard SP-EP and SP-TV approaches since its introduction in 2019 and have included the two extraperitoneal techniques as SP-EP RARP for the current study as both procedures spare the space of the retzius and anterior support structure of the prostate (Figure 1). We currently perform the SP TV approach for RP in very select patients with intermediate-risk prostate cancer as defined in accordance with the National Comprehensive Cancer Network (NCCN) [15], prostate volume less than 80 g, and low risk of pelvic nodal metastasis confirmed by the Briganti nomogram < 7% [8,16]. We also offer this approach in select patients with a hostile abdomen and prior colorectal surgeries with favorable published outcomes [12]. The EP approach is currently reserved for patients with larger prostates (>80 g), higher risk prostate cancer, and/or require standard template pelvic lymph node dissection (PLND) [16].

Briefly, the SP EP-RARP is performed via a single 4 cm midline incision 2 finger breaths below the umbilicus [7], whereas SP TV-RARP is performed via 4 cm midline incision 1–2 finger breaths just above the pubic symphysis with a robot docked directly into the bladder with the patient in a supine or minimal Trendelenburg position in both cases. With regards to SP TV-RARP, insufflation was established only within the confines of the bladder, as aptly referred to as pneumovesicum, following which all of the procedural steps including the dissection and vesicourethral anastomosis can be completed entirely from within the bladder [8,17,18]. The MP-TP RARP is performed in a standard fashion with four robotic arms arranged in a W configuration or in a horizontal line around the umbilicus and the patient in a steep Trendelenburg position as previously published [19,20,21,22].

Baseline patient clinicodemographic variables collected included age, body mass index (BMI), history of previous abdominal surgery, Prostate Specific Antigen (PSA) level, PSA density, and prostate biopsy/pathology, i.e., International Society of Urological Pathology (ISUP) results. Patients who underwent salvage prostatectomy regardless of the surgical platform were excluded from the study. Several perioperative variables were also collected for the study. These include additional procedures performed including lymphadenectomy and nerve-sparing details, the operating time, estimated intraoperative blood loss (EBL), length of stay (LOS), final pathology, surgical margin status, perioperative complications, and readmission. Positive surgical margins (PSM) are defined as either limited or non-limited margins. Limited was defined as a length less than 3 mm, while “non-limited” referred to a length equal to or greater than 3 mm.

Both postoperative complications and readmission were reported within 90 days of surgery. The Clavien–Dindo system was used to classify the postoperative complications with major complications defined as those of grades 3a and above [22]. Urinary continence was defined as complete urinary control using no pads or 1 safety pad in our series. Other functional characteristics collected include patient-specific International Prostate Symptom Score (IPSS) and Sexual Health Inventory for Men (SHIM) scores to evaluate the urinary and erectile functional recovery post-surgery.

### 2.2. Statistical Analysis

A propensity-score matching statistical analysis was performed using the statistical software STATA (Stata Corp LLC, College Station, TX, USA). Descriptive statistics were performed for both groups with categorical variables presented as the absolute and relative percent frequencies, while continuous variables were presented as the mean and 95% confidence interval (CI). A 4:1 matching was conducted based on the ISUP grade group, clinical stage, and preoperative PSA. The significance was set at *p* < 0.05. We currently perform a majority of our RARP via SP in our practice in recent years and for similar indications as MP-RARP and we elected to perform a 4:1 match to include all patients with similar etiology who underwent surgery between 2015 and 2023.

## 3. Results

A total of 925 RARP surgeries were performed within the period. After excluding 26 salvage RARP cases, a propensity-scored analysis was performed on the remaining cases (n = 899). The propensity score-matched analysis, based on the ISUP grade group, clinical stage, and preoperative PSA, resulted in a cohort of 606 patients, of which 121 patients underwent the traditional MP TP approach whereas 485 patients underwent the novel SP EP approach. The SP EP (n = 485) approach included patients who underwent both the traditional EP (n = 259, 53.4%) and TV approaches (n = 226, 46.6%) (Figure 2).

The preoperative patient demographics including mean age (62.6 vs. 63.6 years, *p* = 0.05), BMI (29.1 vs. 29.5, *p* = 0.45), and functional characteristics including SHIM (17.9 vs. 17.6, *p* = 0.63) and IPSS (8.8 vs. 9.3, *p* = 0.5) scores were overall similar between MP and SP patients in the cohort.

Of importance, there was a higher prevalence of prior abdominal surgeries in the SP group compared to MP (41.4% vs. 31.4%, *p* = 0.04). This aligns with staff preference for an EP approach, especially the TV technique in patients with a hostile abdomen to avoid intra-abdominal scar tissue. All SP EP-RARP procedures were completed successfully without the need for conversion or additional ports. The baseline characteristics included in the study are described in detail in Table 1.

### 3.1. Perioperative Outcomes

Regarding the comparison of mean operative times, SP EP-RARP was slightly longer than MP-RARP (198.9 min 95% CI: (195.6–202.2) vs. 181.5 min (95% CI: 174.5–188.6), *p* < 0.001). The SP EP-RARP cases, however, demonstrated lower EBL (125.1 mL, 95% CI 113.5–136.7 vs. 215.9 mL, 95% CI 188.7–243.2) compared to MP.

The SP EP-RARP group demonstrated other significant perioperative benefits including shorter hospital stay [12.6 h (95% CI 10.3–14.8) vs. 31.9 h (95% CI 28.1–35.6), *p* < 0.001], reduced opioid usage upon discharge (14.4% vs. 85.1%, *p* < 0.001), and faster catheter removal [6.0 days (95% CI 5.8–6.3) vs. 8.8 days (95% CI 7.7–9.8), *p* < 0.001].

Regarding the perioperative complications, there was one intraoperative complication in the SP-RARP cohort whereas the postoperative complications were comparable among the groups (15.3% SP vs. 12.4% MP, *p* = 0.38). The intraoperative complication occurred during the TV RARP in a patient with a history of a hostile abdomen. There was a small enterotomy during the initial bladder entry, which was promptly identified and repaired primarily using interrupted Lembert permanent sutures by general surgery.

The overall readmissions were slightly higher with SP (6%) versus MP (3%) in the cohort, *p* = 0.21. The majority of these readmissions in the initial SP cohort were postoperative fluid collections that required percutaneous drainage. Upon clinical and laboratory evaluation of these fluids, they were identified to be postoperative serous collection and not actual lymphocele.

### 3.2. Pathological/Oncological Outcomes

On histopathological analysis, the two groups were identical in terms of the prostatectomy specimen weight, final pathological tumor (T) stage, positive surgical margin (PSM) rate, and overall disease characteristics. From an oncological perspective, the rate of PSM including limited (17.1% vs. 16.5%) and non-limited (8.5% vs. 10%) margins remained comparable across SP vs. MP groups (*p* = 0.84) (Table 2). At a mean follow-up of 13.4 ± 14.3 months, our Kaplan-Meier analysis identified no statistically significant differences in terms of the freedom from biochemical recurrence between the two cohorts (Figure 3).

### 3.3. Functional Outcomes

The mean postoperative follow-up (months) of the SP vs. MP cohort was 12.6 (95% CI 11.6–13.8) vs. 16.3 months (95% CI 12.6–19.9), *p* = 0.013. The urinary continence was assessed at different intervals in our series, i.e., 6 weeks, 3, 6, and 12 months post-surgery. The continence rates were comparable among the two groups in the series at various time points. Urinary continence rates in MP vs. SP groups were 29% vs. 25% at 6 weeks, 56% vs. 53% at 3 months, and 78.5% vs. 80.6% at 6 months post-surgery. The early erectile recovery outcomes, i.e., SHIM scores between SP and MP cohorts at 3 and 6 months post-surgery were equivalent between the two groups, 6.9 vs. 6, *p* = 0.14, and 8.6 vs. 6.7, *p* = 0.13 respectively.

## 4. Discussion

The narrow profile, 360 degree maneuverability, and finer double-jointed surgical instruments with the current SP surgical platform have made it possible to work efficiently in a smaller surgical space without significant instrument clashing compared to the MP platform. SP da Vinci surgical platform has been employed for EP radical prostatectomy at our institution since 2019, and we have performed more than 500 cases of RARP in the last 5 years. We currently perform the SP RARP using the EP and TV approaches.

There was a recent multi-institutional study comparing different SP approaches TP vs. EP by Zeinab et al. [23] in approximately 200 propensity score-matched patients that showed similar advantages of the EP approach. The overall operative times were longer with the EP approach (206 min vs. 155 min, *p* < 0.001). However, the EP approach has shorter hospitalization times (7.5 h vs. 14 h, *p* < 0.001) and a tendency for same-day discharge on short-term follow-up. Our study had similar longer mean operative times with SP EP-RARP compared to MP-RARP (198.9 min [95% CI: 195.6–202.2] vs. 181.5 min (95% CI: 174.5–188.6), *p* < 0.001). The relatively longer operative times with SP are attributable to the initial learning curve, as we included early cases as well in comparison to well-established MP-RARP case times when performed by the same surgeon. The mean LOS impact is even more noticeable in our series when comparing the different approaches (EP vs. TP) with different surgical platforms (SP vs. MP) favoring SP (12.6 h vs. 31.9 h, *p* < 0.001) [23].

Our current study reports that 25% of patients and 53.2% of patients achieved complete urinary continence at 6 weeks and 3 months with SP EP RARP, respectively, and the results were comparable to our MP RARP performed by the same surgeon. Even though the SP and MP cohorts had similar urinary continence rates at various time intervals, we previously published our experience with the TV approach, which, when evaluated individually had superior early continence outcomes after RARP surgery [17]. Immediate continence after foley removal was reported in 49% of cases and early urinary continence was achieved in 65% and 77.4% of patients at 2 weeks and 6 weeks respectively. We believe that early continence was possible with SP TV RARP due to complete sparing of the anterior support structures of the prostate and further study of predictors of early continence in this cohort demonstrated patients with small prostate, low PSA, and low preoperative IPSS scores were the significant predictors for the favorable outcome [24]. A subsequent anatomical study based on the preoperative mpMRI attributed the earlier return of urinary continence in the SP TV RARP cohort to the longer preoperative membranous urethral length, especially with each 1 mm increase in coronal membranous urethral length associated with a 27% increase in the odds of achieving urinary continence at 3 months [25].

Our EP approach had successful perioperative outcomes with a median LOS of 12.6 h, and majority of patients being discharged same-day and with an overall shorter foley catheter duration with a median of 6 days with prior EP and TV surgical approaches. The duration of foley duration in the TV cohort was further reduced with a median time of 3 days as previously described given the EP approach without a significant increase in urinary retention rates [16]. While the decision towards the reduced Foley catheter duration can be dependent on the individual surgeon’s preference, the TV approach remains unique, as it can provide the surgeon the opportunity to evaluate the quality of the vesicourethral anastomosis and the placement of the final Foley catheter from within the bladder, which can thus influence the urinary catheter duration [17]. Despite the favorable perioperative and functional outcomes following the TV approach, a previous study by Abou Zeinab et al. demonstrated comparable oncological outcomes with an identical biochemical recurrence rate following the SP EP and TV techniques [26]. Similar propensity-matched studies involving a lower cohort of patients comparing SP EP-RARP versus MP RARP were conducted by Harrison et al. [27] and Ko et al. [28], which demonstrated similar perioperative outcomes and continence rates between cohorts but overall lower morbidity including decreased opioid use and faster recovery with earlier onset of bowel function post-surgery in the SP EP-RARP cohort likely related to sparing of the peritoneal cavity and the reduced need for intraoperative bowel manipulation.

A recent systematic review by Jiang et al. also highlighted the similarly low morbidity profile between SP TP and SP EP techniques. Nevertheless, when considering major postoperative complications, the incidence was noted to be higher following the EP approach (RR: 0.55, 95% CI 0.31–0.99; *p* < 0.05), particularly pertaining to the incidence of symptomatic fluid collection [29]. This was concordant with our experience whereby the majority of our major postoperative complication was attributed to symptomatic postoperative fluid collection that may necessitate drainage in our initial experience. We have since modified our technique to incorporate a small peritonotomy, i.e., fenestration of the peritoneum at the end of the SP EP-RARP cases at the end of the procedure prior to the fascial closure, which translated to a substantial decrease in its incidence due to the passive drainage of these fluid collections.

When considering the SP TV approach, the possibility of completing the procedure with a supine patient positioning and completely from within the confines of the bladder has allowed for an expanded surgical indication for RARP. This is particularly important when considering whole-gland treatment in patients with a hostile abdomen and/or frozen pelvis, as first reported by Ferguson et al. In the initial series involving 33 patients, most with a maximum hostile abdomen index (HAI) of 4 out of 4 [30], all cases of SP TV RARP were able to be completed successfully without any surgical conversion, bowel injury, or blood transfusion. Postoperative outcomes remained favorable, as indicated by the absence of major postoperative complications and with immediate and 6-week urinary continence achieved in 30% and 51%, respectively [12]. Another example of the expanded indication of RARP was also evident by the possibility of revisiting the utility of epidural anesthesia for RP, as facilitated by the routine supine patient positioning for the more regionalized TV approach. As first described by Kaouk et al., all of the cases were able to be completed without any surgical interruptions and anesthesia-related complications. In addition to providing alternatives to general anesthesia, the utility of epidural anesthesia without mechanical ventilation provided additional opportunities to promote faster postoperative recovery with opioid-sparing outpatient encounters [13].

The current study is not without limitations, the first pertaining to its retrospective nature. It is also a single surgeon and single-institution data. In our current practice, both SP EP-RARP and MP RARP are reserved for unfavorable intermediate-risk, i.e., Gleason grade (GG) 3 prostate cancer (PCa) or higher whereas the SP TV RARP is reserved for intermediate risk (GG 2 or 3) PCa. In addition, when considering the need for PLND, although PLND was routinely performed with the EP approach, a limited LND can also be pursued with the TV approach. This can be completed following the removal of the RP specimen, either by directing the robotic instruments towards the lateral wall and continuing with the dissection towards the obturator lymph nodes, or by withdrawing the inner ring of the SP Access Kit outside of the bladder and completing the PLND using the standard EP approach. In an attempt to limit the selection bias, both cohorts are matched based on the baseline prostate cancer disease characteristics to reduce the confounding factors and include all consecutive SP cases. Despite our propensity-matched analysis, we acknowledge that certain intraoperative factors may contribute to the overall morbidity of the procedure, especially when considering the extent of LND and how they can influence the risk of postoperative lymphocele or fluid collection. Another limitation is the overall intermediate follow-up period. Although there was no difference in biochemical recurrence freedom on the latest follow-up of 13.4 months (SD 14.3), the mean follow-up was 12.6 months in SP vs. 16 months in the MP cohort and more long-term and multi-institutional data are warranted beyond 2 years with SP. Regarding the functional outcomes, the data reported is currently limited to early outcomes and long-term data needs to be reported in future studies.

## 5. Conclusions

Herein, we have demonstrated similar outcomes with SP EP RARP compared to MP TP RARP in terms of oncologic and functional outcomes. However, SP EP RARP offers several advantages in reducing overall hospital stay, decreasing overall opioid use, and shortening the time to catheter removal. This decreases overall patient morbidity and facilitates the transition to outpatient surgery.

## Figures and Tables

**Figure 1 cancers-16-02994-f001:**
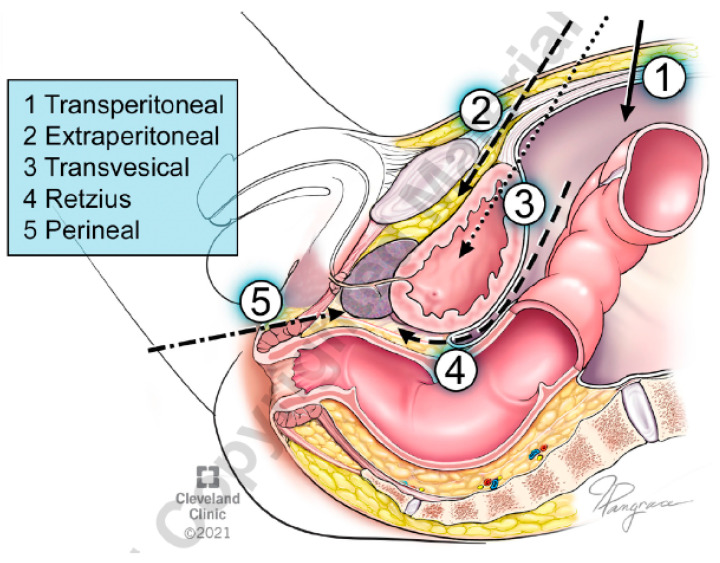
Different surgical access for robot-assisted radical prostatectomy.

**Figure 2 cancers-16-02994-f002:**
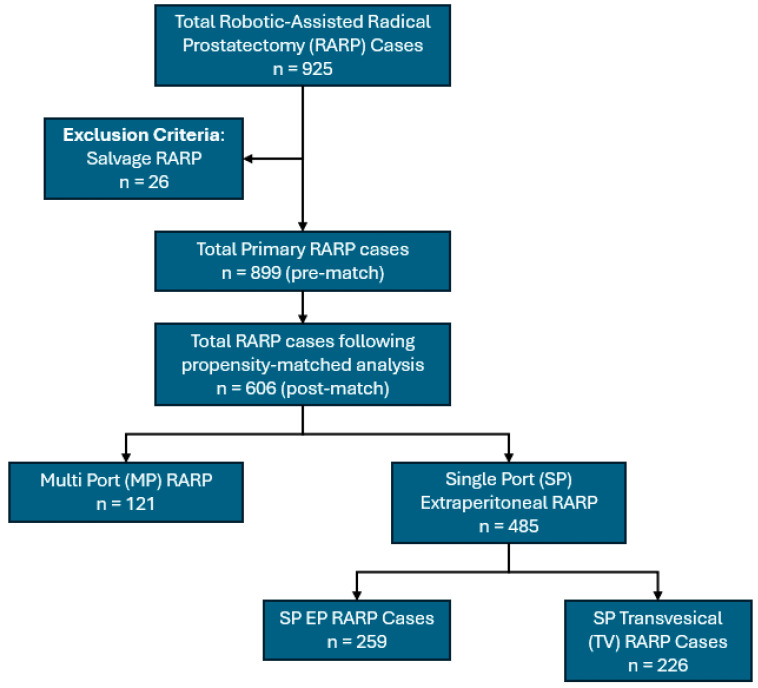
Patient selection criteria for the study.

**Figure 3 cancers-16-02994-f003:**
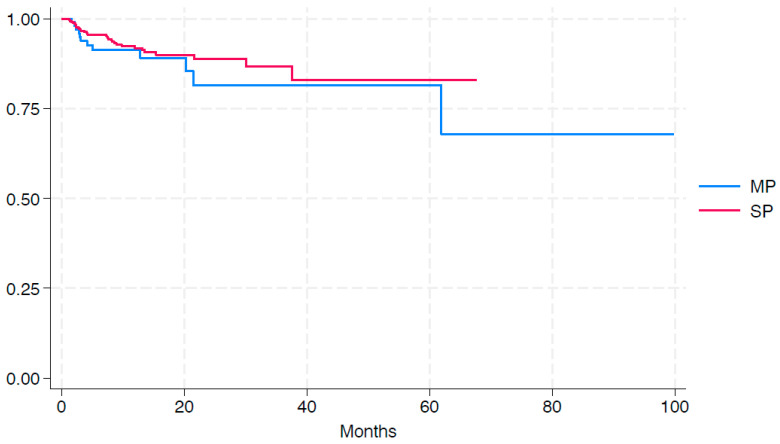
Kaplan-Meier analysis of freedom from biochemical recurrence.

**Table 1 cancers-16-02994-t001:** Patient baseline demographics and perioperative characteristics.

	Before Match	Following Propensity-Matched Analysis		
Parameter	Multiport Transperitoneal n = 429	Multiport Transperitoneal n = 121	Single-Port Extraperitoneal n = 485	*p* Value
Patients Characteristics				
Age, years, mean (95% CI)	61.9 (61.1–62.7)	62.2 (60.9–63.5)	63.6 (63.0–64.2)	0.05
Race, n (%)				0.36
Caucasian	364 (85%)	101 (83.5)	428 (88.2)	
African American	40 (9.3%)	12 (9.9)	33 (6.8)	
Hispanic	12 (2.8%)	8 (6.6)	24 (5.0)	
BMI, kg/m^2^, mean (95% CI)	29.5 (29–29.9)	29.1 (28.3–29.9)	29.5 (29.0–29.9)	0.45
Past surgical history, n (%) ^a^	123 (31.6%)	38 (31.4)	201 (41.4)	**0.043**
**Prostate cancer characteristics**				
PSA, ng/mL, mean (95% CI)	7.3 (6.8–7.8)	8.2 (7.1–9.3)	8.3 (7.6–9.1)	0.87
PSAD, mean (95% CI)	0.24 (0.22–0.26)	0.18 (0.15–0.21)	0.2 (0.19–0.24)	0.13
Percentage of positive cores, mean (95% CI)	39.9 (37.6–42.3)	42.1 (37.6–46.7)	38.5 (36.5–40.5)	0.11
Max core involvement in the core with highest GG, mean (95% CI)	57.1 (54.3–60)	58.8 (52.8–64.8)	57.9 (55.3–60.7)	0.78
ISUP GGG, n (%)				0.25
1	95 (22.1%)	25 (20.7)	87 (17.9)	
2	190 (44.3%)	49 (40.5)	241 (49.7)	
3	73 (17%)	25 (20.7)	95 (19.6)	
4	44 (10.3%)	12 (9.9)	41 (8.5)	
5	27 (6.3%)	10 (8.2)	21 (4.3)	
Clinical stage, n (%)				0.46
T1c	336 (78.3%)	104 (86.0)	408 (84.1)	
T2	51 (11.9%)	16 (13.2)	76 (15.7)	
T3	2 (0.5%)	1 (0.8)	1 (0.2)	
**Functional characteristics**				
SHIM score, mean (95% CI)	18.8 (18.2–19.4)	17.9 (16.6–19.2)	17.6 (16.9–18.2)	0.63
SHIM score ≥ 17, n (%)	230 (72.3%)	71 (58.7)	299 (61.6)	0.86
IPSS, mean (95% CI)	8.7 (8–9.4)	8.8 (7.4–10.2)	9.3 (8.6–10.0)	0.51
**Perioperative characteristics**				
Operative time, min, mean (95% CI)	180 (176.7–183.4)	181.5 (174.5–188.6)	198.9 (195.6–202.2)	<0.001
EBL, mL, mean (95% CI) ^b^	236 (218.7–253.4)	215.9 (188.7–243.2)	125.1 (113.5–136.7)	**<0.001**
Nerve-Sparing, n (%)	352 (83%)	102 (85.0)	415 (85.6)	0.87
Lymphadenectomy, n (%)	386 (90.8%)	110 (90.9)	310 (63.9)	<0.001
Opioids prescribed at discharge, n (%)	374 (88.8%)	103 (85.1)	70 (14.4)	**<0.001**
Length of Stay, hours, mean (95% CI)	34.2 (32.1–36.4)	31.9 (28.1–35.6)	12.6 (10.3–14.8)	**<0.001**
Catheter duration, days, mean (95% CI)	8.6 (8.2–9)	8.8 (7.7–9.8)	6.0 (5.8–6.3)	**<0.001**
Complications, n (%) ^c^				
Intraoperative	0	0	1 (0.2)	0.47
Postoperative	54 (12.8%)	15 (12.4)	75 (15.3)	0.38
Readmission, n (%)	23 (5.4%)	4 (3.3)	30 (6.2)	0.21

BMI: Body Mass Index; PSA: prostate-specific antigen; PSAD: prostate-specific antigen density; ISUP GG: International Society of Urological PathologyGrade Group; SHIM: Sexual Health Inventory for Men; IPSS: International prostate symptom score; EBL: Estimated Blood Loss. ^a^ Past Abdominal Surgical History included the following: Umbilical, Inguinal, or Ventral hernia repairs w/wo mesh, appendectomy, proctocolectomy with ileostomy, land colectomy. ^b^ Only one patient in the multiport group required blood transfusion. ^c^ Postoperative complications were recorded within 90-days of the respective surgery. The bold refers to statistically significant variables.

**Table 2 cancers-16-02994-t002:** Pathological, oncological, and functional outcomes.

Pathologic Outcomes	Multiport Transperitoneal n = 121	Single-Port Extraperitoneal n = 485	*p* Value
Prostate weight, g, mean (95% CI))	56.2 (50.0–62.4)	53.7 (51.0–56.4)	0.42
Pathologic T stage, n (%)			0.45
T2	55 (45.5)	239 (49.3)	
T3a/b	66 (54.5)	246 (50.7)	
Pathologic Lymph Node Stage, n (%)			0.64
Nx	11 (9.1)	175 (36.1)	
N0	102 (84.3)	291 (60.0)	
N1	8 (6.6)	19 (3.9)	
ISUP GG, n (%)			0.30
1	7 (5.8)	25 (5.1)	
2	70 (57.8)	312 (64.3)	
3	25 (20.7)	104 (21.5)	
4	5 (4.1)	11 (2.3)	
5	14 (11.6)	33 (6.8)	
Surgical margin details, n (%) ^a^			0.84
Negative	89 (73.5)	361 (74.4)	
Limited	20 (16.5)	83 (17.1)	
Non-limited	12 (10.0)	41 (8.5)	
**Oncologic outcomes**			
Freedom from BCR at 60 mos., mean (95% CI) ^b^	81.7 (81.6–81.8)	83.0 (82.9–83.1)	0.36
Follow-up, months, mean (95% CI)	16.3 (12.6–19.9)	12.6(11.6–13.8)	0.013
**Functional outcomes**			
SHIM, mean (95% CI)			
6 weeks	5.0 (4.8–5.3)	6.0 (5.6–6.5)	0.03
3 months	6.0 (5.0–6.9)	6.9 (6.3–7.6)	0.14
6 months	6.7 (4.5–9.0)	8.6 (7.6–9.6)	0.13
0–1 safety pad/day, n (%)			
6 weeks	35 (28.9)	121 (24.9)	0.37
3 months	68 (56.2)	258 (53.2)	0.55
6 months	95 (78.5)	391 (80.6)	0.60
12 months	110 (91.1)	138 (90)	0.59

ISUP GG: International Society of Urological Pathology Grade Group; BCR: Biochemical recurrence. ^a^ “Limited” was defined as a length less than 3 mm, while “non-limited” referred to a length equal to or greater than 3 mm. ^b^ See Figure 1.

## Data Availability

We would like to confirm that the data presented in this study are available on request from the corresponding authors per the patient confidentiality statement and regulation set forth by our institution and the Health Insurance Portability and Accountability Act (HIPAA).

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
