# Peer review of "Single-Port Extraperitoneal vs. Multiport Transperitoneal Robot-Assisted Radical Prostatectomy: A Propensity Score-Matched Analysis"

_cancers, 2024, doi:10.3390/cancers16172994_

Round 1

Reviewer 1 Report

Comments and Suggestions for Authors

This retrospective comparative study assesses the impact of extraperitoneal single-port robotic prostatectomy with the standard transperitoneal multiport approach. The study is handicapped by its retrospective, single-surgeon nature. However, it could be considered for publication should the authors address the following: 

1) Methods (lines 82-88): The authors should better define the technique selection criteria regarding tumor risk. More specifically, they should define the intermediate-high risk status and remove the "higher risk" status. What happened to patients with high-risk tumors where extended lymph node dissection was necessary? Was a transperitoneal approach not performed in this case? How did this influence the flow diagram?

2)  Results: Perioperative outcomes (lines 150-151). The authors should replace the term ''slightly'' as the difference is statistically significant. 

3) Results: Perioperative outcomes (lines 150-151) and Discussion (lines 223-226). The authors should explain possible reasons for significantly less catheterization duration in the intervention group. This parameter could be influenced by several factors not related to the technique itself (e.g., surgeon experience). Furthermore, the authors should explain the criteria for catheter removal (e.g., empiric preference according to the case). Was the catheter removal time influenced by imaging findings like retrograde cystography?

4) Results: Pathological/oncological outcomes. Strangely, the T stage did not differ significantly between the two groups, as the transperitoneal group should also include high-risk locally advanced (>T2c) patients. The authors should explain this in the discussion-limitations. 

5) Results: Functional outcomes. The authors should include 12-month continence outcomes regarding urinary continence as their follow-up was longer than one year.  

6) Citations: I understand that the surgeon is a pioneer in this matter. However, the authors should modify the discussion to reduce self-citations to a minimum significantly

Comments on the Quality of English Language

Needs polishing by a language-editing software 

Author Response

This retrospective comparative study assesses the impact of extraperitoneal single-port robotic prostatectomy with the standard transperitoneal multiport approach. The study is handicapped by its retrospective, single-surgeon nature. However, it could be considered for publication should the authors address the following: 

1) Methods (lines 82-88): The authors should better define the technique selection criteria regarding tumor risk. More specifically, they should define the intermediate-high risk status and remove the "higher risk" status. What happened to patients with high-risk tumors where extended lymph node dissection was necessary? Was a transperitoneal approach not performed in this case? How did this influence the flow diagram?

Author's Response: Thank you for your comment. The specific prostate cancer risk was defined in accordance with the National Comprehensive Cancer Network (NCCN) criteria. In addition, we routinely used the Briganti nomogram to evaluate the risk for lymph node dissection, with patients who scored above 7% commonly referred for lymph node dissection. Of note, extended pelvic lymph node dissection can be safely performed using the extraperitoneal approach. The manuscript has been amended accordingly to include the aforementioned points. The flow diagram did not pertain to our patient selection criteria for the individual approach, rather it relates to the patient inclusion for this particular study and the relevant analysis.

2)  Results: Perioperative outcomes (lines 150-151). The authors should replace the term ''slightly'' as the difference is statistically significant. 

Author's Response: Thank you for your feedback. We have removed the word per your suggestion.

3) Results: Perioperative outcomes (lines 150-151) and Discussion (lines 223-226). The authors should explain possible reasons for significantly less catheterization duration in the intervention group. This parameter could be influenced by several factors not related to the technique itself (e.g., surgeon experience). Furthermore, the authors should explain the criteria for catheter removal (e.g., empiric preference according to the case). Was the catheter removal time influenced by imaging findings like retrograde cystography?

Author's Response: Thank you for your comment and we agree with the reviewer that there are different factors that can contribute to the decision towards reduced Foley catheter duration. These can include surgeon preference and the evaluation of the vesicourethral anastomosis. For the Transvesical approach in particular, the latter remains important, especially given the ability to completely visualize each suture placement and the final Foley catheter placement from within the bladder. Retrograde cystography was not routinely performed as our standard of care.

4) Results: Pathological/oncological outcomes. Strangely, the T stage did not differ significantly between the two groups, as the transperitoneal group should also include high-risk locally advanced (>T2c) patients. The authors should explain this in the discussion limitations. 

Author's Response: Thank you for your question. The similar T stage on histopathology was likely influenced by our statistical analysis method whereby we adopted a propensity-matched analysis to reduce the selection bias when comparing between the SP and MP approaches. The two cohorts were matched based on their preoperative ISUP groups, PSA level, as well as clinical stage.

5) Results: Functional outcomes. The authors should include 12-month continence outcomes regarding urinary continence as their follow-up was longer than one year.  

Author's Response: Thank you for your comment. We have included the 12-month continence outcomes based on your feedback.

6) Citations: I understand that the surgeon is a pioneer in this matter. However, the authors should modify the discussion to reduce self-citations to a minimum significantly. 

Author's Response: Thank you for your suggestion. We appreciated your comments and have included additional references accordingly.

Reviewer 2 Report

Comments and Suggestions for Authors

Comments to the author

General comments

This paper is a comparative study of SP-RARP and MP-RARP, and is a very significant report because of the large number of cases involved. There are minor concerns that I would suggest.

Specific comments

Minor

1)     Among the SP-RARP, 259 SP EP RARP and 226 SP TV RARP were included.

The Methods section described patient selection methods, but SP EP RARP and SP TV RARP are two very different procedures, and I wonder if there is any difference in biochemical recurrence or pathological outcome between SP EP RARP and SP TV RARP. The comments should be added to the Discussion section.

2)     In particular, SP TV RARP opens the apex of the bladder, so it is debatable

whether the duration of catheter placement should be shortened.

Author Response

1) Among the SP-RARP, 259 SP EP RARP and 226 SP TV RARP were included. The Methods section described patient selection methods, but SP EP RARP and SP TV RARP are two very different procedures, and I wonder if there is any difference in biochemical recurrence or pathological outcome between SP EP RARP and SP TV RARP. The comments should be added to the Discussion section.

Author's Response: Thank you for your question. We have previously performed a propensity-matched analysis based on the baseline prostate cancer disease characteristics to evaluate whether there are differences in the perioperative and oncological outcomes between the SP Transvesical and Extraperitoneal approaches where we did not identify any significant differences in the biochemical recurrence following the two techniques (Abou Zeinab M, Beksac AT, Ferguson E, Kaviani A, Kaouk J. Transvesical versus extraperitoneal single-port robotic radical prostatectomy: a matched-pair analysis. World J Urol. 2022 Aug;40(8):2001-2008. doi: 10.1007/s00345-022-04056-6. Epub 2022 Jun 19.) We have included the relevant changes in our discussion per your suggestion.

2) In particular, SP TV RARP opens the apex of the bladder, so it is debatable whether the duration of catheter placement should be shortened.

Author's Response: Thank you for your comment and we agree with the reviewer that there are different factors that can contribute to the decision towards reduced Foley catheter duration. These can include surgeon preference and the evaluation of the vesicourethral anastomosis. For the Transvesical approach in particular, the latter remains important, especially given the ability to completely visualize each suture placement and the final Foley catheter placement from within the bladder.  

Reviewer 3 Report

Comments and Suggestions for Authors

This is an interesting paper comparing two robotic approaches for prostatectomy. One of them is no so widely implemented so all the information shared is relevant. However, the authors have already published about it and the research presents some issues that should be addressed

  • Only 20 more patients than a previous similar publication from the group. It should be notified and further explain the added value of this new publication

·        Why IMC, age and Prostate volume were not included as variables for the propensity analysis? Both of them could influence in surgical complications and outcomes more than PSA for example.

  • Results should be compared and discuss with recent publish systematic review 10.1186/s12957-023-03272-7
  • Not only the figure but a table showing the changes in the descriptive analysis pre and post propensity analysis between groups should be shown.
  • Lymphadenectomy percentage was quite a lot higher in the transperitoneal approach vs extraperitoneal. This could have influence results in terms of EBL and pain. Why the authors have not adjusted by this variable?
  • A comment about the double rate of readmission in extraperitoneal approach should be added despite not being significant.
  • Did the surgeon put a drain in any of these patients?

Author Response

1) This is an interesting paper comparing two robotic approaches for prostatectomy. One of them is no so widely implemented so all the information shared is relevant. However, the authors have already published about it and the research presents some issues that should be addressed. Only 20 more patients than a previous similar publication from the group. It should be notified and further explain the added value of this new publication

Author's Response: Thank you for your review and valuable feedback. The value of this paper is not to report the feasibility of the SP Extraperitoneal approach, but rather to compare the perioperative, oncological, and functional outcomes of the SP approach with the gold-standard Multi-Port Robotic Transperitoneal approach. To our knowledge, this study represents the first to perform such a comparison.  

2) Why IMC, age and Prostate volume were not included as variables for the propensity analysis? Both of them could influence in surgical complications and outcomes more than PSA for example.

Author's Response: Thank you for your question. Given our objective was to include all consecutive series of the SP Extraperitoneal cohort and our primary objective was to report the safety and oncological outcomes, we only included the baseline prostate cancer disease characteristics in the propensity-matched analysis. However, we did identify that age and final prostate specimen weight remain comparable between the two approaches.

3) Results should be compared and discuss with recent publish systematic review 10.1186/s12957-023-03272-7

Author's Response: Thank you for your suggestion. We have included an additional comment in our discussion pertaining to the highlighted systematic review. Nevertheless, it is important to note that the systematic review compared the surgical outcomes of two Single Port approaches (Extraperitoneal and Transperitoneal) while our study sought to compare SP Extraperitoneal with the traditional MP Transperitoneal approach.

4) Not only the figure but a table showing the changes in the descriptive analysis pre and post propensity analysis between groups should be shown.

Author's Response: Thank you and we have included the numbers for the MP cohort prior to the propensity-matched analysis in Table 1 per your suggestion.

5) Lymphadenectomy percentage was quite a lot higher in the transperitoneal approach vs extraperitoneal. This could have influence results in terms of EBL and pain. Why the authors have not adjusted by this variable?

Author's Response: Thank you for your suggestion. We only considered preoperative variables in our propensity-matched analysis and hence we did not account for the intraoperative factors. We appreciate your suggestion and we may consider this in future studies to include a larger cohort of patients and to better evaluate the longer term oncological outcomes, including for the lymph node yield and how they impact on the biochemical recurrence and distant metastases.  

6) A comment about the double rate of readmission in extraperitoneal approach should be added despite not being significant.

Author's Response: Thank you for your comment. We agree with the reviewer regarding the importance of this finding and we appreciate the opportunity to clarify. Most cases of readmission following the SP Extraperitoneal approach occurred in our earlier experience, which was related to symptomatic postoperative fluid collection, some of which necessitated percutaneous drainage. We have since adopted a routine peritonotomy at the end of the procedure prior to fascial closure, which has since reduced the incidence of symptomatic postoperative fluid collection (line 252).

7) Did the surgeon put a drain in any of these patients?

Author's Response: Thank you for your question. Our standard of practice did not include the placement of a surgical drain.

Round 2

Reviewer 1 Report

Comments and Suggestions for Authors

The authors addressed my comments adequately. No further comments 

Author Response

The authors addressed my comments adequately. No further comments 

Author Response: Thank you for your overall feedback, comments, and suggestions. 

Reviewer 3 Report

Comments and Suggestions for Authors

Thank you for your responses, some of them should be added and at least explain in the paper as limitations:

As described in the methodology and as the authors results shown lymph node dissection is a presurgery plan approach independently of the number of nodes removed and clearly could affect the results in terms of complications so the differences in the rate of Lymphadenectomies between both techniques could be affected the results. This should be included as limitations.

The authors answered “We have since adopted a routine peritonotomy at the end of the procedure prior to fascial closure, which has since reduced the incidence of symptomatic postoperative fluid collection (line 252).” About the higher rate of readmission, but in the current discussion section there is no comment about it, please move it from the result to the discussion section.  Furthermore, they mention the following “The higher incidence of major postoperative complication following SP TP approach compared to SP  EP technique was also demonstrated by a recent review by Jiang et al.(28)” This current study does not find any differences between both approaches but this initial tendency of higher readmission rates until technique modification. Please clarify in the discussion section.

Other comments:

This sentence “Although PLND was routinely 105 performed with the EP approach, a limited LND can also be pursued with the TV ap- 106 proach. This can be completed following the removal of the RP specimen, either by directing the robotic instruments towards the lateral wall and continuing with the dissection towards the obturator lymph nodes, or by withdrawing the inner ring of the SP Access Kit  outside of the bladder and completing the PLND using the standard EP approach.” Should be moved to discussion as it is not part of your methodology.

Author Response

Thank you for your responses, some of them should be added and at least explain in the paper as limitations:

As described in the methodology and as the authors results shown lymph node dissection is a presurgery plan approach independently of the number of nodes removed and clearly could affect the results in terms of complications so the differences in the rate of Lymphadenectomies between both techniques could be affected the results. This should be included as limitations.

Author Response: Thank you for your feedback. We have included this comment in our limitations section per your suggestion.

The authors answered “We have since adopted a routine peritonotomy at the end of the procedure prior to fascial closure, which has since reduced the incidence of symptomatic postoperative fluid collection (line 252).” About the higher rate of readmission, but in the current discussion section there is no comment about it, please move it from the result to the discussion section.  Furthermore, they mention the following “The higher incidence of major postoperative complication following SP TP approach compared to SP  EP technique was also demonstrated by a recent review by Jiang et al.(28)” This current study does not find any differences between both approaches but this initial tendency of higher readmission rates until technique modification. Please clarify in the discussion section.

Author Response: Thank you for your comment and we apologize for the confusion. We have amended our manuscript to include further clarification and modification on the relevant sections. 

Other comments:

This sentence “Although PLND was routinely 105 performed with the EP approach, a limited LND can also be pursued with the TV ap- 106 proach. This can be completed following the removal of the RP specimen, either by directing the robotic instruments towards the lateral wall and continuing with the dissection towards the obturator lymph nodes, or by withdrawing the inner ring of the SP Access Kit  outside of the bladder and completing the PLND using the standard EP approach.” Should be moved to discussion as it is not part of your methodology.

Author Response: Thank you for your suggestion. We agreed with the reviewer and have modified our manuscript as per your feedback. On behalf of the co-authors, we thank you for your overall review, comments, and feedback. 

Round 3

Reviewer 3 Report

Comments and Suggestions for Authors

Thank you. All comments addressed